



# Observations of strong turbulence and mixing impacting water exchange between two basins in the Baltic Sea

Julia Muchowski[1], Martin Jakobsson[1], Lars Umlauf[2], Lars Arneborg[3], Bo Gustafsson[1,4], Peter Holtermann[2], Christoph Humborg[1,4], Christian Stranne[1,4]

[1]Department of Geological Sciences, Stockholm University, Stockholm, 10691, Sweden
[2]Institute for Baltic Sea Research, Warnemunde, 18119, Germany
[3]Department of Research and Development, Swedish Meteorological and Hydrological Institute, Gothenburg, 426 71 Sweden
[4]Baltic Sea Centre, Stockholm, 10691, Sweden

*Correspondence to*: Julia Muchowski (julia.muchowski@geo.su.se)

**Abstract**

Turbulent diapycnal mixing is important for the estuarine circulation between basins of the Baltic Sea as well as for its local ecosystems, in particular with regard to eutrophication and anoxic conditions. While the interior of the basins is overall relatively calm, stratified flow over steep bathymetric features is known as a source for strong turbulent mixing. Yet, current

in situ observations often cannot capture dynamic and intermittent turbulent mixing related to overflow over rough bathymetry. We present observational oceanographic data together with openly accessible high-resolution bathymetry from a prototypical sill and an adjacent deep channel in the sparsely-sampled Southern Quark located in the Åland Sea, connecting the Northern Baltic Proper with the Bothnian Sea. Our data include high resolution broadband acoustic observations of turbulent mixing, in situ microstructure profiler measurements, and current velocities from Acoustic Doppler Current Profilers and were acquired

during two one-week cruises in February-March 2019 and 202. A temporally reversing non-tidal stratified flow over the steep bathymetric sill created a dynamic and extremely energetic environment. Saltier, warmer, and less oxygenated deep water south of the sill was partly blocked, the reversing flow was at times hydraulically controlled with hydraulic jumps occurring on both sides of the sill, and sub-mesoscale processes in the surface layer leading to high spatial variability at small scales. Mixing and vertical salt flux rates were increased by 3-4 orders of magnitude in the entire water column in the vicinity of the

sill compared to reference stations not directly influenced by the overflow. We suggest based on acoustic observations and in situ measurements that underlying mechanisms for the highly increased mixing across the halocline are a combination of shear and topographic lee waves which are breaking at the halocline interface. We anticipate that the resulting deep- and surface-water modification in the Southern Quark directly impacts exchange processes between the Bothnian Sea and the Northern Baltic Proper and that the observed mixing is likely important for oxygen and nutrient conditions in the Bothnian Sea. Our

results contribute to the knowledge on turbulent mixing processes in the Åland Sea and can help to improve mixing parametrizations in numerical models of the area.





**Plain Language Summary**

We show observational data of highly increased mixing and vertical salt flux rates in a sparsely sampled region of the Northern
Baltic Sea. Co-located acoustic observations complement the in-situ measurements and visualize turbulent mixing with high
spatial resolution. The observed mixing is generally not resolved in numerical models of the area but likely impacts the
exchange of water between the adjacent basins as well as nutrient and oxygen conditions in the Bothnian Sea.

**1 Introduction**

The Southern Quark is the northern part of the Åland Sea, linking the Northern Baltic Proper with the Bothnian Sea through
the deepest connection between the two basins (Fig. 1). Water mass properties, including salinity and temperature, differ
greatly between the Northern Baltic Proper and the Bothnian Sea. While the deep water of the Baltic Proper has become
increasingly anoxic due to eutrophication and poor ventilation (Carstensen et al., 2014), the deep water in the Gulf of Bothnia,
which includes the Bothnian Sea and the Bothnian Bay north of it, is currently well oxygenated.

Predictions of future oxygen conditions in the Bothnian Sea are contradictory. While overall oxygen conditions in the oceans
may further exacerbate in response to climate warming (Keeling et al., 2010; Breitburg et al., 2018; Schmidtko et al., 2017),
the situation in the Bothnian Sea is less clear. On the one hand, a study based on long-term monitoring data from 1960-2015
(Raateoja, 2013) and a comparison of six different Baltic Sea models (Meier et al., 2018) predict that the Bothnian Sea will
likely remain well oxygenated; the latter study even suggests that its stratification will decrease despite increased warming.
On the other hand, an analysis of long-term monitoring data from 1980-2015 (Kuosa et al., 2017) and a high resolution model
of the Baltic Sea combined with several downscaled global climate projections (Gröger et al., 2019) point to enhanced
stratification from increased runoff and warming of surface waters, aligned with decreased oxygenation. The link between
stratification and oxygen conditions in the Bothnian Sea has also been investigated in paleo-oceanographic studies. Jilbert et
al. (2015) suggest that the isostatic uplift since the last glacial maximum lead to gradually shallower sill depths in the Åland
Sea and consequently less salty water entering the Bothnian Sea. Their data show that the resulting weaker stratification lead
to well-ventilated bottom waters in the Bothnian Sea that prior to this suffered from anoxic episodes.

Although stratification often does play an important role for oxygenation, it has been shown that the amplification of anoxic
conditions in the Baltic Sea during the last century is in fact mainly attributed to eutrophication (Kuliński et al., 2022; Meier
et al., 2018). Even though nutrient loads from the Baltic states have decreased since 1980, nutrient loads from previous decades
have accumulated in the sediments and are released into the water in anaerobic conditions (Gustafsson et al., 2012). This effect
leads to a continuation of increased nutrient concentrations in the water despite that anthropogenic, land-based sources have
declined significantly. Potentially related to this long-term feedback mechanism, it has been shown that phosphorus
concentrations in the Bothnian Sea have increased in the last decades and are now posing a risk of increasingly anoxic



conditions in the currently overall well oxygenated basin. As a consequence of the increased phosphorous concentrations, cyanobacteria have started to bloom in the Bothnian Sea (Olofsson et al., 2021). Rolff & Elfwing (2015) suggest that the observed increase in phosphorus originates from the Northern Baltic Proper and that it is being transported northward through the Åland Sea into the Bothnian Sea. This suggests that a sound understanding of mixing and water mass transformations in the topographically complex Åland Sea region is essential for predicting the evolution of physical and ecosystem parameters

in the Bothnian Sea.

The underlying causes for observed trends in oxygen conditions in the Bothnian Sea are not fully understood, but it is well known that turbulent vertical mixing and water exchange processes in the Åland Sea affect the physical and chemical conditions in the Bothnian Sea, and in fact the entire Gulf of Bothnia (Hela, 1958; Palosuo, 1964; Marmefelt and Omstedt,

1993; Westerlund et al., 2022). By altering the stratification in the basins, turbulent vertical mixing is also important for the larger-scale circulation. Yet, inflow and transformation of deep water from the Northern Baltic Proper through the Åland Sea into the Bothnian Sea, have rarely been studied and are still poorly understood. Hietala et al. 2007 state that "the paucity of observational data which could be used for direct comparisons is immediately striking", referring to the Åland Sea. This is still true today, and the lack of in situ observational data from the Southern Quark (the northern part of the Åland Sea, Fig. 1b)

hampers the development and verification of ocean models in the area. Knowledge about temporal and spatial variations of the vertical salt flux and the underlying turbulent mixing is needed to develop models and improve mixing parametrizations (Meier et al., 2006). Oceanographic processes in the area are highly influenced by the particularly variable bathymetry. Only recently, bathymetric data were made openly accessible through EMODnet (EMODnet Bathymetry Consortium, 2020), beginning with a resolution of 225x225 m in 2014, and with increased resolution of 115x115 m since 2018.


Here, we present observational data from the Southern Quark in the northern Åland Sea, including densely-spaced observations of velocities, stratification, and mixing parameters, and data from a broadband echosounding system that provide insights into the complex spatial structure of turbulent mixing at unprecedented resolution. We compare our measurements with previously published observational data as well as model output from a recently published high-resolution numerical model of the Åland

Sea (Westerlund et al. 2022). We discuss the impact of the observed turbulent mixing on the temperature and salinity stratification of the Åland Sea deep-water, which eventually propagates northward into the Bothnian Sea.





## 2. Background

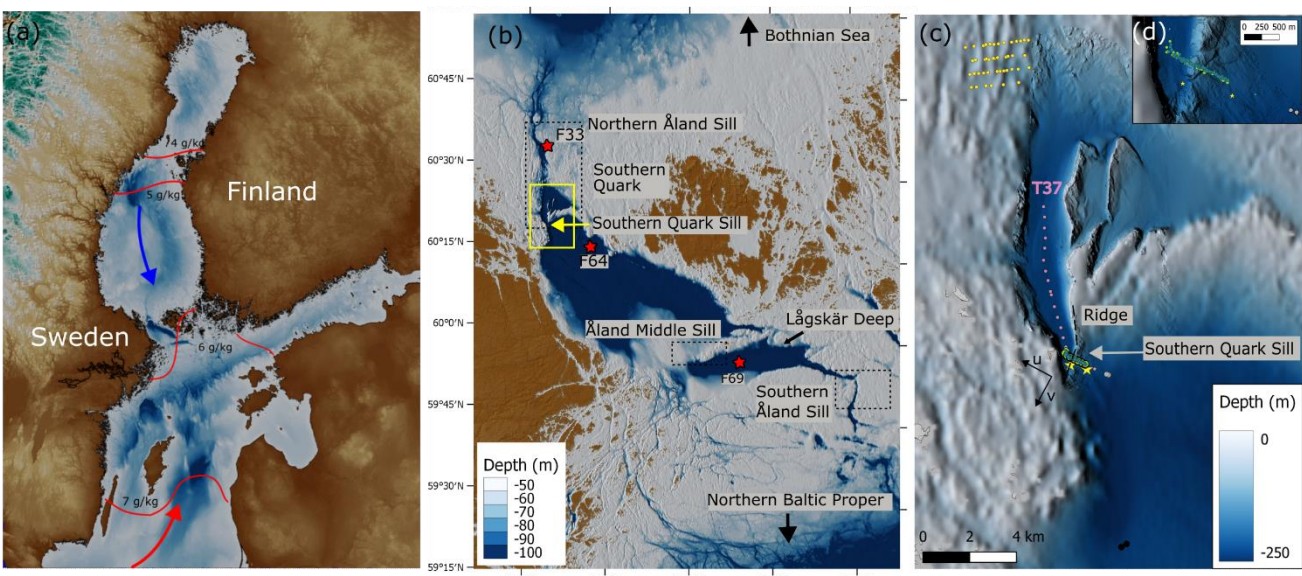

**Figure 1: (a) Northern Baltic Sea with approximate position of surface isohalines (red lines) from (Snoeijs-Leijonmalm and Andrén, 2017), main flow direction of saltier deep water entering the Baltic Sea from the North Sea (red arrow) and fresh surface water from northern Baltic Sea (blue arrow). (b) Overview over study area as shown in Westerlund et al. (2022) with three major sills that separate the Northern Baltic Proper from the Bothnian Sea (black dashed rectangles), positions of previous observations from monitoring stations (red stars). Yellow rectangle marks region of data collection within this study shown in (c). Bathymetric data from EMODnet (EMODnet Bathymetry Consortium, 2020). (c) Positions of MSS casts and mooring ADCPs during cruises EL19-IGV01 Feb 2019 and EL20-IGV01 Feb-March 2020. MSS profiles near Southern Quark Sill during EL19 (turquoise) and EL20 (green), reference stations south of sill during EL19 (black) and EL20 (grey), transect T37 during EL19 (rose), parallel transects in northwestern part of the study region collected during EL20 (yellow). ADCP moorings (yellow stars) - EL19-IGV01 only north of sill and EL20-IGV01 at both positions, south and north of sill. (d) Southern Quark Sill bathymetry enlarged. Background bathymetry data from EMODnet (EMODnet Bathymetry Consortium, 2020), higher resolution multibeam bathymetry data were acquired by R/V Electra and granted public release by the Swedish Maritime Administration (release- 17-03187).**

### 2.1 Bathymetry of the study area

The Åland Sea bathymetry is among the most complex in the Baltic Sea (Jakobsson et al., 2019) (Fig. 1b). While the western and southern parts host deep basins and steep ridges, the eastern and northern areas are characterized by shallow seafloors, with a few generally less than 30 m deep channels cutting through the Åland Archipelago. The Lågskär Deep is located south of Åland and has depths exceeding 150 m. The basin's deepest connection to the Northern Baltic Proper is in the south through a bathymetric channel with a sill depth of about 60 m according to the gridded EMODnet Digital Bathymetric Model (DBM) (EMODnet Bathymetry Consortium, 2020). Following Westerlund et al. (2022) and Leppäranta and Myrberg (2009), we adopt the name "Southern Åland Sill" for this sill and "Åland Middle Sill" for the passage at the western end of Lågskär Deep (Fig.



1b). The latter sill also has a depth of about 60 m and is the deepest connection northward to the Southern Åland Sea Basin, which is the deepest part of the Åland Sea with a maximum depth of 295 m according to the EMODnet grid. The northern part of the Southern Åland Sea Basin extends into the Bothnian Sea, if the definitions by the Baltic Marine Environment Protection

Commission (HELCOM) are applied (Bergström and Avellan, 2018). This area of the north-western Åland Sea and southern Bothnian Sea forms a deep-water passage between Åland and the Swedish mainland commonly referred to as the Southern Quark (Fig. 1). There is a ~88 m deep sill in the Southern Quark located in a bathymetric channel linking the Southern Åland Sea Basin and the deeper Bothnian Sea (Fig. 1b). This sill is here referred to as the "Northern Åland Sill", which is consistent with Westerlund et al. (2022).


The seafloor of our study area in the Southern Quark is highly dramatic with a major ridge running in east-west direction nearly across the entire passage. This ridge has a crest that rises up to a depth of ~25 m and which ends with a steep cliff towards the flat seafloor to the north where the water depths are generally deeper than 150 m. In contrast, the slope of the ridge facing southward is gentle. The deepest passage over the ridge is a ~ 170-m deep sill at its southern edge with narrow channels

that are up to 200 m deep. We established a transect across this sill, slightly northeast of this deepest passage (Fig. 1c), and will refer to it as the "Southern Quark Sill". All sills described above are most likely crucial for the exchange flow between the sub-basins. It is therefore important to understand their role in blocking and mixing of water masses. We consider the Southern Quark Sill as a prototypical example for the region, giving rise to the most relevant turbulent mixing processes.

The dramatic seafloor morphology of the Southern Quark is to a large extent inherited from the underlying bedrock geology. The Southern Åland Sea basin and Lågskär Deep were formed from a tectonic depression and are mainly underlain by Jotnian sandstone of Mesoproterozoic age (1.0-1.6 Ga), while the rough seafloor areas surrounding Åland as well as the islands themselves, are predominantly comprised of older crystalline bedrock, i.e. to a large extent the famous Rapakivi granite (EMODnet Geology) (Beckholmen and Tiren, 2009). The steep ridge in the Southern Quark mentioned above is proposed to

be composed of post-Jotnian dolerite, which is a magmatic rock formed through injection of a fluid commonly into sedimentary rocks (Beckholmen and Tirén, 2009).

Jakobsson et al. (2019) showed that the EMODnet DBM (EMODnet Bathymetry Consortium, 2020) is capable of capturing the main characteristics of the dramatic Southern Quark seafloor morphology, while the older lower resolution DBM

IOWTOPO, compiled by the Leibniz-Institut für Ostseeforschung Warnemünde (IOW) (Seifert et al., 2001, 1995), is smoothing over critical details. Yet, many previous studies have made use of IOWTOPO (Tuomi et al., 2012, 2018; Lessin et al., 2014; Dargahi et al., 2017; Meier et al., 2003). The grid cell-size of IOWTOPO2 covering the Southern Quark area is 2 × 1 arc-minutes (longitude × latitude), which is ~1.9 × 1.9 km at 60°N. The EMODnet DBM has a resolution of 1/16 × 1/16 arc-minutes equating to ~58 × 116 m at the same latitude. In addition, EMODnet has a considerably larger underlying source



database. The above highlights the importance of considering the DBM when trying to understand processes that influence water circulation and water mass transformation, e.g. in ocean circulation models, and in particular for a region such as the Southern Quark with its extreme bathymetry on small scales.

**2.2 Oceanography of the study area**

Characteristic for the Baltic Sea is its pronounced large-scale horizontal north-south salinity and temperature gradient (Snoeijs-Leijonmalm and Andrén, 2017; Leppäranta and Myrberg, 2009; Kullenberg, 1982) (Fig. 1a). Saline deep waters from the North Sea enter the Baltic Sea through the Kattegat in the south and flow northward along the bottom topography of the Baltic Sea basins, while large volumes of fresh water enter the Baltic Sea at the surface in the form of river runoff, precipitation and ice melt, especially in its northern parts (Kullenberg, 1982) (Fig. 1a). In most sufficiently deep regions of the Baltic Sea, the

fresh surface water is separated from the saltier deep water by a pronounced halocline that persists throughout the year at a depth of around 60-80 m (Ahola, 2021). A general estuarine-type circulation occurs all the way to the northernmost parts of the Baltic Sea in the Gulf of Finland (Elken et al., 2003) and Gulf of Bothnia (Marmefelt and Omstedt, 1993; Green et al., 2006).

Our study area lies between the Baltic Proper and the Bothnian Sea in the southern part of the Gulf of Bothnia. Observations and numerical modelling results of the average bottom (surface) water salinity vary between 7.5-11 g kg$^{-1}$ (6-7 g kg$^{-1}$) in the Baltic Proper and between 5-7.5 g kg$^{-1}$ (4-6 g kg$^{-1}$) in the Bothnian Sea (Snoeijs-Leijonmalm and Andrén, 2017; Westerlund and Tuomi, 2016; Kullenberg, 1982; Raateoja, 2013). Overall, salinity decreases towards the north. The mean surface temperature lays between 7-9 °C in the Baltic Proper and between 5-7 °C in the Bothnian Sea, as estimated from numerical

modelling (Kniebusch, 2019). In the Northern Baltic Proper, the permanent strong halocline exists all year at varying depth of around 60-80 m (Väli et al., 2013) as in most parts of the Baltic Sea, while in the Bothnian Sea, a comparably weak halocline can be found at varying depths between 40-80 m (Håkansson et al., 1996; Westerlund and Tuomi, 2016; Kullenberg, 1982, p.198). In some parts of the Bothnian Sea, the halocline may even be absent in winter, but stratification is stable throughout the year (Kullenberg, 1982).


Differences between water properties in the Baltic Proper and the Bothnian Sea are to a large part due to the sills and channels in the Åland Sea described above (section 2.1, Fig. 1) that partly block the flow of saline deep water from the Northern Baltic Proper into the Bothnian Sea. Previous studies describe inflow into the Bothnian Sea as one component of a larger-scale estuarine circulation. Saline waters from below the surface layer down to a depth of 50-70 m propagate northward, following

the seafloor topography. The deep water flows from the Northern Baltic Proper through small channels and canyons over the Southern Åland Sill and through the Åland Sea towards the Bothnian Sea (Hela, 1958; Håkansson et al., 1996; Leppäranta and Myrberg, 2009). At the same time, an approximately 15 m deep thermocline develops in the Bothnian Sea (Håkansson et al.,



1996) and the fresher surface layer flows southward into the Northern Baltic Proper (Hela, 1958; Marmefelt and Omstedt, 1993; Håkansson et al., 1996; Myrberg and Andrejev, 2006; Leppäranta and Myrberg, 2009). The southward surface layer flow along the Swedish coast is strong with velocities up to 5-9 cm s$^{-1}$ (Myrberg and Andrejev, 2006). During winter, oxygen-rich surface water in the Northern Baltic Proper is at times forced by strong winds northward into the Åland Sea (Kullenberg, 1982; Palosuo, 1964). Strong southerly/south-westerly winds are known to temporarily reverse the general estuarine circulation in the Northern Baltic Proper and Gulf of Finland, causing a compensating southward flow of the deep water (Elken et al., 2003).

Turbulent mixing is an essential component of any type of estuarine circulation as it is the main process for water mass transformation (Geyer and MacCready, 2014; Burchard et al., 2018). This is also true for estuarine circulation in the northern Baltic Sea, where mixing in the Åland Sea has been shown to dilute and modify northward flowing water, which eventually becomes the deep water of the Bothnian Sea (Hela, 1958; Palosuo, 1964; Kullenberg, 1982; Neumann et al., 2020), as well as southward flowing water, which becomes part of the surface water in the Northern Baltic Proper (Hela, 1958; Håkanson and Bryhn, 2008; Markus Meier et al., 2006). The effect of mixing on the estuarine overturning circulation in this complex transition region, however, has not been studied in detail, mainly due to the lack of reliable mixing data and problems of numerical models to provide a precise and robust representation of mixing. The water renewal time in the Bothnian Sea has been estimated to one year by Kullenberg (1982) and to four to five years by Håkansson et al. (1996). In support of a great variation in renewal time, Meier (2005) found that the age of the bottom water in the Bothnian Sea varies largely with a median of 1.8 years and a maximum age of 4.2 years. These differences may be related to temporal changes, differences in the definitions, and uncertainties when modelling exchange processes along the transport pathways between the Baltic Proper and the Bothnian Sea.

## 3. Methods

Data presented in this study were collected during two cruises with R/V *Electra*: EL19-IGV01, 21 February – 26 February 2019, and EL20-IGV01, 27 February – 6 March 2020. These cruises are henceforth referred to as EL19 and EL20 respectively. During both cruises measurements were acquired repeatedly along a transect across the Southern Quark Sill in the southern part of the study region (Fig. 1c). During EL19, this transect was extended further north along the valley (Fig. 1c, rose dots). An area that was not visited during EL19, but extensively surveyed during EL20 was located in the north-western corner of the study region where we collected measurements along four parallel transects (Fig. 1c) to understand mixing in a region with extremely corrugated topography (Muchowski et al., 2022b under review).

Acoustic observations were collected continuously with the hull-mounted EK70-7C split beam transducer installed on RV *Electra*. This transducer has a 7° circular beam width and is connected to a Simrad EK80 wideband transceiver. A 4.1-ms long



chirp pulse ranging from 45 to 90 kHz at a transmit power of 750 W and a ping rate of 1 s was used. Match-filtering of the chirped pulse leads to a vertical resolution of about 1.5 cm. The horizontal resolution varies with the ship speed. The system was calibrated in the study area during both cruises. During the EL19 cruise, a 38.1 mm tungsten carbide sphere was used while a 32 mm copper sphere was used during EL20. We display acoustic backscatter strength per volume $S_v$ in dB relative to a reference of 1 μPa. Additional information about the acoustic system as well as a detailed analysis how energy dissipation

rates from turbulent diapycnal mixing can be inferred from the acoustic broadband observations can be found in Muchowski et al. (2022).

The acoustic observations were combined with co-located in-situ measurements from a free-falling Sea & Sun Technology MSS90-L microstructure profiler (MSS). Exact times and positions of all MSS casts collected during EL19 and EL20 are listed

in Supplementary Tables ST1 and ST2, respectively. The MSS profiler was equipped with two PNS06 airfoil shear probe sensors, an internal shear sensor, a FP07 fast thermistors, and precision CTD sensors, including an oxygen sensor. The falling speed of the profiler was adjusted to approximately 0.7 m s$^{-1}$. From shear sensor measurements, dissipation rates of turbulent kinetic energy ε were obtained. See Muchowski et al. 2022 for detailed information on the MSS hardware components as well as processing steps. Based on the energy dissipation rates and the buoyancy frequency $N$, turbulent diffusivities $k_z$ were

calculated using (Osborn, 1980)

$$k_z = \frac{\gamma\varepsilon}{N^2} \, , \tag{1}$$

thereby assuming a constant flux coefficient γ = 0.2 (Gregg et al., 2018). Vertical salinity flux rates were calculated from the diffusivity $k_z$ using Fick's law

$$F_{zS} = -k_z \cdot \frac{\partial S}{\partial z} \cdot \rho \, , \tag{2}$$


with water depth $z$ defined positive upward, water density $\rho$ and salinity $S$.

Current velocities were measured with R/V *Electra's* hull-mounted Teledyne 600 kHz Workhorse ADCP, providing reliable velocity data down to approximately 40 m depth. Additionally, we deployed moored upward looking 300 kHz Workhorse

ADCPs during both cruises. Data from the hull-mounted ship ADCP were averaged over 60 seconds. During EL19, one ADCP mooring was deployed at 215 m depth on 22 February 2019 and recovered on 27 February 2019 at 60°16'20.21''*N*, 18° 55'48.25''*E* (Fig. 2a). During EL20-IGV01 two ADCP moorings were deployed. ADCP 1 was deployed on 28 February 2020 and recovered on 6 March 2020 at approximately the same position as the ADCP mooring deployed during EL19. ADCP 2 was deployed at 190 m depth on 27 February 2020 and recovered on 6 March 2020 at 60° 15' 55.8'' *N,* 18° 56' 38.399''*E*

(Fig. 1c). The moored ADCPs sampled the water column every second in 2-m bins and during post processing data were averaged to 1-min intervals for noise reduction. All data were processed with an IOW in-house software package that includes a quality control algorithm. Detailed information can be found in Muchowski et al. 2022.





## 4. Results and Discussion

In the first part, we discuss a single transect, measured across the Southern Quark Sill from south to north and through the
adjacent valley further north. The impacts of topographic blocking and mixing on the properties of the water north of the sill
are examined – in the surface water, across the halocline and in the deep water. The Southern Quark Sill and the valley
connected to it can be seen as one example out of a variety of bathymetric features in the Åland Sea that modify the water
properties and therefore impact the exchange flow between the adjacent basins.

In the second part, we show temporal variations of the flow over the Southern Quark Sill and estimate the amount of mixing
in the vicinity of the sill on time scales of days to weeks. In order to do so, all MSS casts collected near the sill during both
cruises are included and compared to reference stations south of the sill as well as to previously published measurements in a
shallower area northwest of the sill (Fig. 1c).

### 4.1 Water modification along one particular transect

### 4.1.1 Microstructure measurements

During transect T37 (Table ST1), measured on 26 February 2019 between 14:50 and 17:40 across the Southern Quark Sill
(Fig. 1c), we collected in total 19 MSS casts, of which 17 cover the full water depth and are included in the following analysis.
Profiles of absolute salinity and conservative temperature show fresh, cold surface water above warmer, saltier bottom water
separated by a halo- and thermocline at 20-50 m depth (Fig 2c,d). The simultaneously collected ship ADCP data reveal that at
the beginning of the almost 3-h long transect, the surface layer flow in the vicinity of the sill was southward, as expected for
an estuarine-type circulation, while north of the sill, the flow direction changed towards the south (at distance of about 4.5 km
from the start of the transect) after 1.5 hours into the transect (Fig. 2a,b). Corresponding wind data from SMHI station Örskär
(60.5256 °N, 18.3729 °E) in the vicinity of the study region reveal a shift in wind direction and drop in wind speed during the
time of the measurement that coincides with the change in surface water current direction (Supplementary Fig. S1).


ADCP data (Fig. 3) from the mooring slightly north of the sill show that, during the time of the measurement, the main flow
direction of the deep-water below 140 m depth was northwest and northward below the halocline but above 140 m (the near-
surface layer is outside the range of this instrument). Note that data collected along the transect show spatial and temporal
variations. The deep water had a flow velocity of about 0.25 m s$^{-1}$ near the sill and based on this, it would travel ~2.7 km during
the approximately three-hour duration of the 8-km long transect.

The deepest warm and salty water with lowest oxygen concentrations south of the sill is partly blocked from propagating
northwards (Fig. 2c-e). Particularly striking are the strongly enhanced energy dissipation rates in the wake of the sill, reaching
several kilometres, or $O(10^1-10^2)$ sill heights, into the valley (Fig. 2f). In this region, deep-water isopycnals show regular





undulations with a wavelength of approximately 1500 m (assuming stationarity) and vertical excursions of up to 50 m, which are also reflected in the structure of salinity and oxygen, e.g. between 2-3 km and 4-5 km distance along the transect (Fig. 2c,d). We interpret this pattern as the signature of stationary lee waves generated at the sill, triggering deep-water turbulence and mixing in the wake of the sill. A direct consequence of the highly increased mixing north of the sill is the slight freshening and cooling of the deep water from south to north that can be identified in (Fig. 2c,d). The same trend is visible also in the

oxygen data, where oxygen concentrations in the deep water increase from south to north.

Especially important for the estuarine circulation are the enhanced dissipation rates in the halocline region (30-50 m depth), where vertical temperature and salinity gradients are strong and turbulence will thus effectively mix inflowing and outflowing waters. We will see in the following that turbulence in this region is also clearly reflected in our acoustic turbulence

measurements.



**Figure 2: Measurements collected along transect T37 shown in Figure 1c. (a) Ship ADCP northward velocity, (b) ship ADCP westward velocity, both averaged over 60 s. (c)-(f) Interpolation of MSS 212-230, excluding casts 221 and 224 which were aborted. (c) Conservative temperature, (d) absolute salinity, (e) oxygen concentration, (f) energy dissipation rate. Black isopycnals plotted at intervals of 0.05 g kg⁻¹, grey isopycnals at 0.01 g kg⁻¹. In panel (d) and (f) black isopycnals in the halocline are plotted at intervals of 0.2 g kg⁻¹. Exact time and position of all MSS casts collected during EL19 are shown in Supplementary Table ST1.**





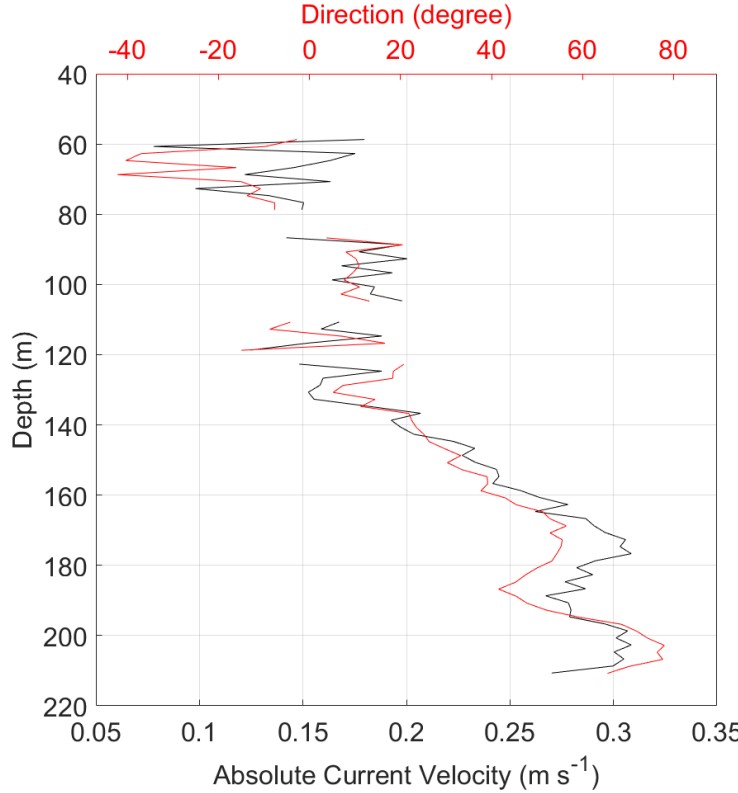

**Figure 3: Average velocity during the time of transect T37 as measured with moored upward-looking ADCP slightly north of the Southern Quark Sill. Data rotated by 90 degree as in Fig. 2a,b, 0 degree are equivalent to northward direction. Validation parameter data can be found in the Supplementary Material Fig. S3.**



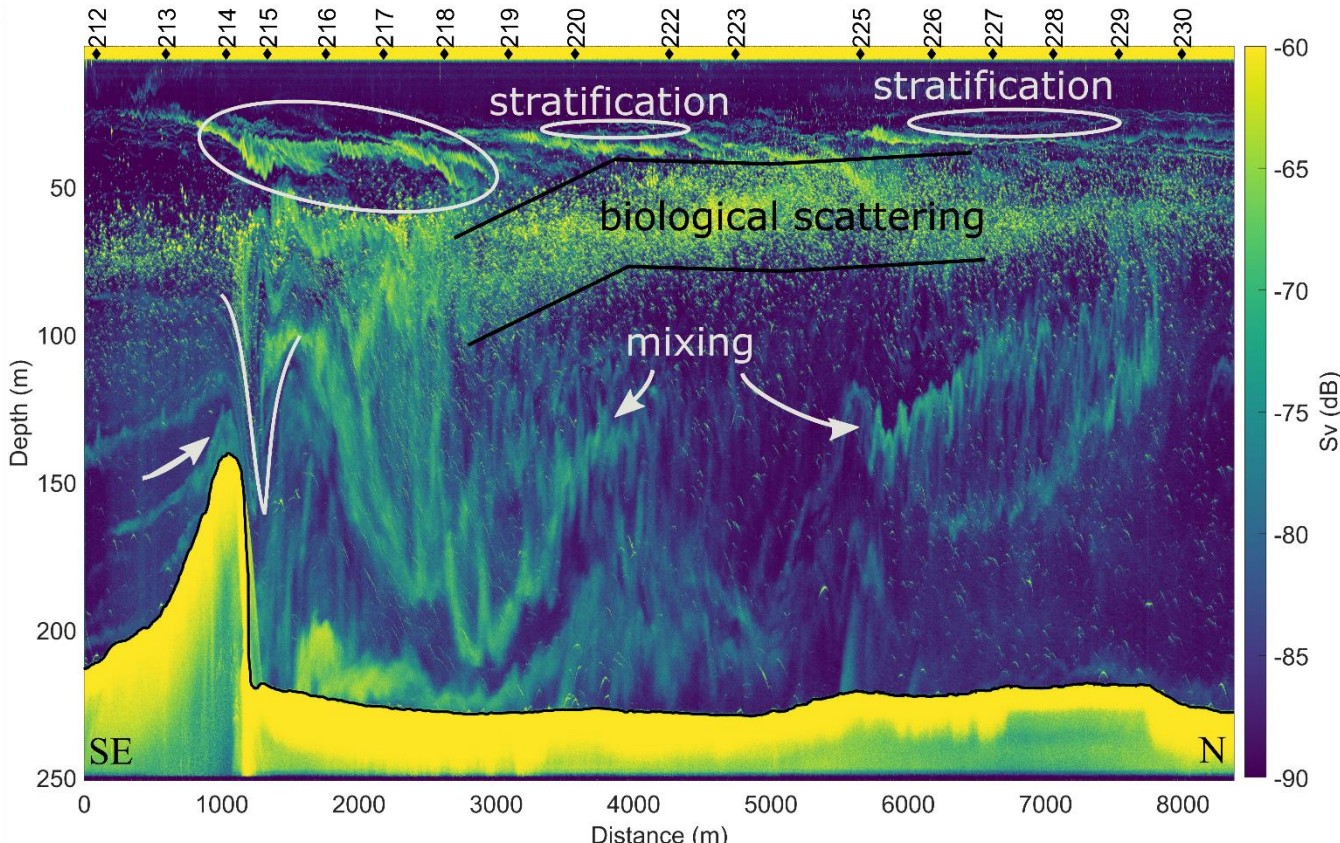


**Figure 4:** Echogram collected on transect T37 during EL19 while crossing the Southern Quark Sill and passing the valley north of it. Acoustic observations are co-located with MSS casts 212-230 shown in Fig. 2 (black diamonds) and given as volume backscatter strength in dB. Black line outlines the seafloor bathymetry. Water flowing from left (southeast) to right (north) over the sill and further north through the valley (Fig. 1c). Highly increased vertical mixing in the halocline region particularly strong just north of
the sill, with small-scale oscillations (bright grey ellipsoid starting above the sill crest), as well as large patches of increased turbulent mixing on the lee side of the sill below the halocline (bright grey arrows). Signs of backscatter from stratification in the halocline (two narrow bright grey ellipsoids) as well as bands of increased backscatter below the sill level that are tilted upwards just south of the sill (bright grey arrow). Distinct scatterers are likely biological targets, such as fish or zooplankton (in region between black lines).


### 4.1.2 Acoustic Observations

We collected broadband acoustic data (Fig. 4) co-located with the microstructure measurements. In Muchowski et al. 2022, a clear correlation between increased acoustic backscatter and regions of increased mixing in this dataset was found, and it was shown that turbulent mixing rates can reliably be estimated from acoustic backscatter strength in combination with temperature
and salinity profiles in regions where the acoustic backscatter is not dominated by biological scattering. The acoustic observations of turbulent mixing (Fig. 4) correspond excellently with in-situ measurements of energy dissipation (Fig. 2f). While the microstructure profiler data only provide snapshots at a few positions with a vertical resolution of 0.5 m, the acoustic



observations show the detailed structure of the turbulence in the sill region with a vertical resolution of about 2 cm and a horizontal resolution of about 1 m (Fig. 4). One can see, due to eroding and tilting density interfaces, how the deep water is
lifted up south of the sill (bright grey arrow in Fig. 4), and falls down on the lee side of the sill where biological single target scatterers become thin stretched-out lines (indicating that they are transported passively with the current). The sill is thus causing highly increased turbulent mixing as indicated in the echogram by blurry, cloudy backscatter without clearly defined edges. Remnants of a stagnant, wedge-shaped pool of water (e.g. Farmer and Armi, 1998; Cummins et al., 2006) are visible just north of the sill at a depth of around 50-100 m (V-shape in Fig.4), pointing towards conditions in which hydraulic control
is or was recently present. Acoustic backscatter from diapycnal turbulent mixing is visible in direct connection to the overflow north of the sill and extends for more than four kilometres downstream in the entire water column below the surface mixed layer, perfectly consistent with the shear-microstructure data discussed above. In a layer between 40-100 m depth, distinct strong biological scattering masks acoustic observations of turbulent mixing, complicating the identification of turbulence microstructure in this region (Muchowski et al. 2022). In the deeper regions below 100 m, however, large overturns, eddies,
and internal-wave oscillations are ubiquitous in the echogram (Fig.4) as well as in the microstructure data (Fig. 2c-e), causing increased mixing all the way down to the seafloor (Fig. 2f). Further away from the sill, increased mixing rates in the deep water are likely also influenced by the steep walls on both sides of the valley.

While the acoustic observations are insensitive to increased dissipation rates in the nearly well-mixed surface layer, where
there is little temperature and salinity microstructure, their sensitivity is highest in stratified regions, such as in the halocline (Muchowski et al., 2022). Indeed, strong scattering from turbulent diapycnal mixing is visible in the halocline region along the entire transect, again similar to the microstructure profiler data discussed above. Particularly strong turbulent diapycnal mixing can be seen below the surface mixed layer and above approximately 50 m depth as tilted bands with small-scale oscillations (Fig. 4) that may be signs of Kelvin Helmholtz instabilities (Geyer et al., 2010). The observed strong diapycnal
mixing across the halocline is particularly important for the exchange flow between the basins. It is likely crucial for the water mass properties of the outflowing surface layer, as it entrains salt, as well as for the outflowing deep-water which becomes diluted and enriched with oxygen. Without high-resolution current measurements inside and below the halocline, we can only speculate that the enhanced mixing in the halocline region is due to (a) mean current shear across the halocline associated with the near-surface freshwater outflow and (b) breaking of lee waves generated at the sill in the strongly sheared and stratified
halocline, thereby depositing energy from the overflowing deep water in the halocline (Fig 2a,b and Fig. 3). The tilted overall pattern supports the lee wave breaking hypothesis.

In some parts of the echogram (Fig. 4), signs of additional backscatter from stratification (Lavery and Ross, 2007; Stranne et al., 2017, 2018) can be seen as thin distinct lines just above the strong fuzzy scattering bands from turbulent mixing.
Backscatter from stratification is high for interfaces that are sharp compared to the wavelength of the acoustic wave and is expected to be relatively weak in the study environment as temperatures only increase by about 3° over a depth range of about





20 m, without many visible interfaces. For instance, scattering analysed in Lavery and Ross (2007) was due to double diffusion and included changes of 6 ° over a depth of only 4 cm. Nevertheless, acoustic backscatter from stratification will likely contribute to the total backscatter signal in the halocline, and is likely not negligible under calm conditions, especially in

regions with large changes in temperature over small depth ranges.

### 4.1.3 Temperature and Salinity Profiles

Arranging temperature and salinity profiles by region shows that the deep water south of the sill (MSS 212-214) is about 1 degree warmer and almost 0.2 g kg$^{-1}$ saltier than the water in the valley further north of the sill (MSS 229,230). This is in

agreement with the above mentioned partial blocking of the saline and warm deep water south of the Southern Quark Sill. The observed tendencies of a freshening surface mixed layer and a sinking halocline depth towards the north (Fig. 5c, MSS casts 212-228) are in agreement with previous observations (e.g. Hela, 1958) and the concept of an estuarine-type circulation.

However, our data also suggest a strong small-scale intermittency of mixed layer depth and near-surface stratification: in agreement with tilted isopycnals in the surface layer (Fig. 2c,d), the two northernmost MSS profiles (MSS 229,230) show a

significantly different signal in the upper 50 m than all other profiles collected on the transect (Fig. 5a and c). MSS profiles 229 and 230 show a well-mixed surface layer extending down to nearly 20 m, while MSS profile 228 observed only 9 minutes earlier and only 470 m south, shows no clear evidence for a surface mixed layer at all (Fig. 5c). The measured change in ADCP surface currents described above (over time and in space) could explain the unexpected temperature and salinity signal in the upper 50 m of the two MSS measurements furthest north (Fig. 5f, dark blue dots). We can only speculate that for instance

wind and/or shallow bathymetric structures east and west of the transect could have altered the surface current direction in the northernmost part of the transect. From a high-resolution modelling study, Chrysagi et al. (2021) recently argued that mixed-layer re-stratification due to sub-mesoscale frontal instability is ubiquitous in the central Baltic Sea, and it is likely that our data show evidence for this process also in more northern parts of the Baltic Sea.



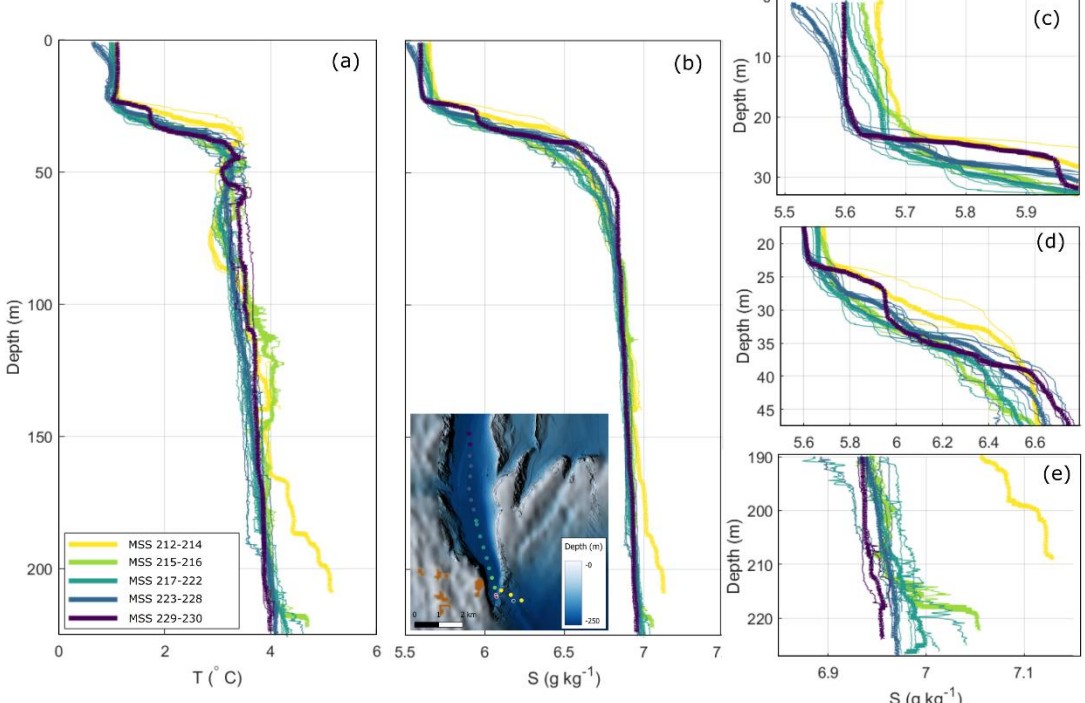

**Figure 5: MSS profiles of conservative temperature (a) and absolute salinity (b) at positions shown on map (Figure 2). (c) –(e) show absolute salinity in mixed layer, halocline and bottom layer as in (b). (f) map of collected data along transect across the Southern Quark Sill. Dots represent MSS casts 212 to 230, collected during EL19, on 26 Feb 2019, 14:54-17:43 UTC, shown in (a)-(e). Deep water salinity and temperature decrease from south to north. Halocline becomes deeper towards the north. The two profiles furthest north show a distinctly different signal than the others.**

## 4.2 Time series of mixing in the vicinity of the Southern Quark Sill

In order to capture the temporal variability and average mixing rates related to the Southern Quark Sill, the southern part of the transect shown in Figures 2-5 was sampled repeatedly during both cruises EL19 and EL20. Observational data were collected along a 2-3-km long transect across the Southern Quark Sill. Broadband acoustic observations of this sampling effort, backed up by microstructure measurements, reveal the temporal development of the flow at high spatial resolution. We interpolated acoustic data from subsequent transects across the sill and created in total three videos showing how flow direction and mixing rates change over time. Continuous sampling of the sill took place during EL19 between 25 Feb. 2019, 18:30 UTC and 26 Feb. 2019, 16:00 UTC (Video SV1, Table ST1, Supplementary) and during EL20 between 4 March 2020, 21:30 UTC and 5 March 2020, 14:30 UTC as well as between 5 March 2020, 17:20 UTC and 6 March 2020, the latter sampling took place without collocated MSS measurements (Video SV2, Table ST2, and Video SV3, Supplementary). During all continuous measurements of the transect, the upward looking ADCP moorings were deployed and the collected data are shown in Figures 6 and 7. The ADCP time series show that the deep-water flow across the Southern Quark Sill was not always northwards, as



the case during the single transect shown in Figures 2-5. In fact, flow reversals in the deep water occurred during both cruises, EL19 and EL20, on the time scales of days (Figs. 6 and 7), which is in good agreement with the acoustic observations (SV1-

SV3) and MSS data (not shown here in the form of interpolated transects) that show increased turbulent mixing alternating south and north of the sill. During EL19, the acoustic observations show how hydraulic control is established, leading to trapped internal waves on the lee side of the sill that cannot propagate upstream anymore (Video SV1). At about 6:00 UTC on 26 Feb 2019, hydraulic control breaks down, causing increased mixing in the entire water column and one can see how shear instabilities and internal waves move upstream again. After a short phase of calmer conditions, deep water currents pick up

again towards the end of the video and mixing is increased again. During EL20, acoustic backscatter from turbulent mixing was overall weaker but one can clearly see how the flow direction changes repeatedly in both videos, causing increased backscatter from turbulent mixing on either side of the sill (SV2 and SV3).

While absolute current velocities were comparable during both cruises, the direction of the flow differs between the two

datasets. During EL19, a flow reversal occurred only once, from the dominant northward direction to southward flowing currents on the second day of the cruise. During EL20, flow reversals occurred more frequently and the direction of the deep-water flow was around 60 % of the time northward. While we only deployed one moored ADCP north of the sill in 2019, we deployed two moored ADCPs south and north of the sill in 2020. Assuming an average stratification in the water below the halocline of $N^2 = 10^{-5}$ s$^{-2}$ and a sill height of $h = 80$ m, we calculated the topographic Froude number $F_r = u/(Nh)$, using the

absolute velocity as measured by the moored ADCPs, averaged from 150 m water depth to the bottom, for southward and northward flow. The estimate of $N^2$ creates uncertainty in the following discussion, as it can easily vary and fluctuate by one order of magnitude over time and in space. Nevertheless, with the definition above, a Froude number of one indicates that there is likely a hydraulic jump at the position of the mooring, a Froude number above one (higher velocities than $F_r =1$) indicates that conditions for a hydraulic jump are likely met in the vicinity of the ADCP measurement. During EL19, $F_r$ was

above one 38 % of the time for northward flow at the position of the ADCP mooring north of the sill. In EL20 the condition was never met for northward flow at the position of the mooring north of the sill but it was met 32 % of the duration of the deployment for southward flow at the position of the mooring south of the sill. This might explain the energy dissipation and vertical flux rates in the region around the sill shown below (Fig. 8), which were comparable during the two cruises although northward flow in the deep layer was much weaker during EL20 than it was during EL19. Overall currents north of the sill

were stronger and more persistent during EL19 (compare Fig. 6 with Fig. 7a).

The predominant non-tidal estuarine-type circulation with deep water flowing northward, is often reversed in the presented ADCP observations. Periods with increased measured sea level at the SMHI station in Forsmark (slightly north of the study area, 60.4086 °N, 18.2108 °E) compared to the sea level at the SMHI station in Landsort Norra (south of the study area,

58.7689 °N, 17.8589 °E), are consistently followed by reversing, southward flow in the deep water (Figs. 6 and 7). Wind velocity and direction as measured at SMHI station Örskär in the vicinity of the study region during cruise EL19



(Supplementary Fig. S1) and cruise EL20 (Supplementary Fig. S2) change on time scales that are comparable to those at which current directions in the deep water change and vary more frequently during EL20 compared to EL19. Based on changes in sea level and wind data, reversing flow conditions are likely caused by pressure gradients set up by atmospheric forcing. The

observed flow reversals occur on time scales of days and our data suggest that the deep water flushes back and forth between the basins in the Åland Sea. Flow reversals occurred more frequently during EL20 than during EL19. Yet, deep water flow reversals occurred during both cruises on time scales of several days, in contrast to previously estimated flow reversals on time scales of several months (Kullenberg 1981).

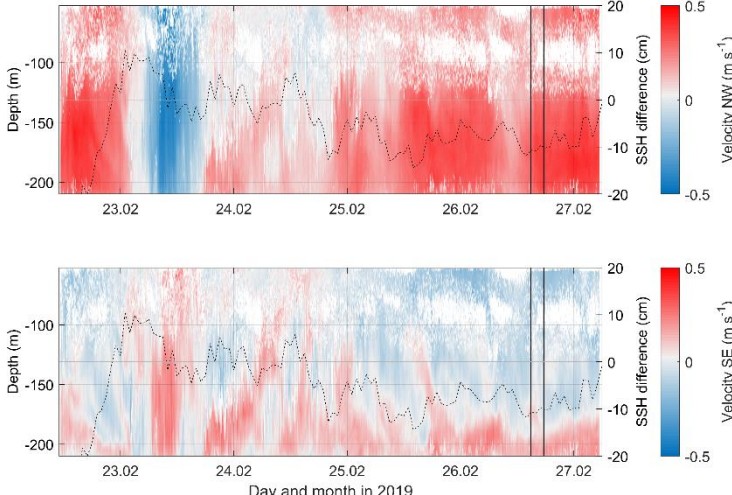

**Figure 6: ADCP mooring data collected during EL19 north of sill. Vertical black lines mark start and end times of the transect shown in Fig. 2 and 3. Position of the mooring and flow directions are shown in Fig. 1c, directions are aligned with the repeatedly sampled transect across the Southern Quark Sill. Dotted lines show difference in sea surface height between Forsmark (slightly north of the study region) and Landsort (south of the study region) from SMHI.**



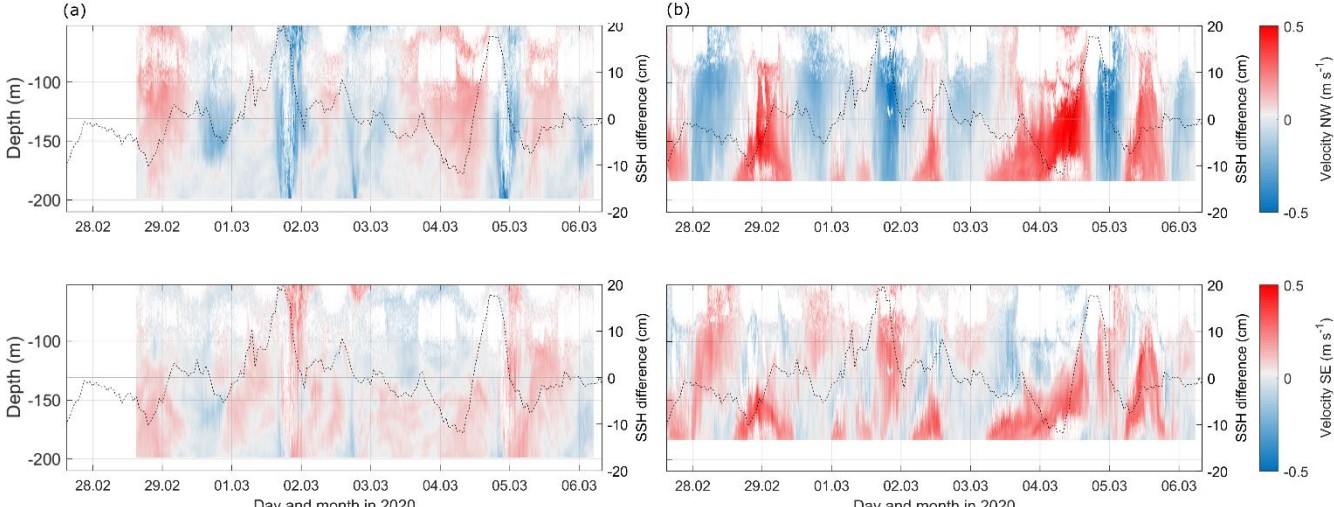

**Figure 7:** ADCP mooring data collected during EL20 (a) north of sill, and (b) EL20 south of sill (positions of moorings and directions along and across the transect are shown in Fig 1c). Dotted lines show difference in sea surface height between Forsmark (slightly north of the study region) and Landsort (south of the study region) from SMHI.





**Figure 8: Reference stations south of the sill in EL19 (MSS 3,5) and EL20 (MSS 18,19,152) (black and grey, respectively), 99 MSS profiles near sill in EL19 (turquoise), 64 MSS profiles near sill in EL20 (green), stations in northwest shallow area (yellow) and their average (thick lines). Dates and exact position of MSS casts used are listed in table ST3 (casts during EL19) and ST4 (casts during EL20), Supplementary. The halocline region is marked as shaded background with corresponding colours.**

To quantify the amount of mixing related to the reversing overflow of deep water over the Southern Quark Sill during the entire time of both cruises, we compare all MSS profiles collected in the vicinity of the sill with reference stations south of the sill (Fig. 8). During EL19, MSS casts were obtained several kilometres south of the sill and are here used as a reference station which is far enough away to be unaffected by the overflow over the Southern Quark Sill (Fig. 1c, black dots). During EL20 such full-depth MSS casts were unfortunately not collected. With the lack of a better option we therefore use MSS casts



obtained about 500 m south of the southernmost MSS casts that are on the transect chosen for the analysis (Fig. 1c,d, grey dots). We find that energy dissipation, turbulent vertical mixing and vertical salt flux rates are increased by three to four orders

of magnitude near the Southern Quark Sill compared to reference stations south of the sill, during both cruises EL19 and EL20 (Fig. 8d-f). Additionally, as shown in Muchowski et al. (under review), diapycnal mixing across the halocline was increased up to two orders of magnitude in a north-western part of the study region during EL20 (Fig. 8, yellow lines).

A high-resolution (500 m) modelling study of the Åland Sea was recently published by Westerlund et al. (2022), using the 3D

hydrodynamic model NEMO 4.0.3. They show that previous CTD measurements of the deep water at the entrance of the Bothnian Sea are on average about 0.5 g kg$^{-1}$ fresher and about 1 °C colder than modelled by NEMO (their figure 3). While NEMO overestimates the deep-water salinity, it underestimates the surface water salinity on average by 0.5 g kg$^{-1}$ compared to CTD measurements, in both the Åland and the Bothnian seas.  Furthermore, the deep water in the Lågskär Deep, south of the Åland Sea Proper but north of the southern Åland Sill (station F69 in their figure 1b), is observed to be on average saltier

than predicted by NEMO. While observations show a change in average deep-water salinity from above 8 g kg$^{-1}$ (station F69) to 7 g kg$^{-1}$ in the Southern Quark (F64) to 6.5 g kg$^{-1}$ in the Bothnian Sea (F33), the deep-water salinity in NEMO only changes from just below 8 g kg$^{-1}$ south of Southern Quark to just above 7 g kg$^{-1}$ in the Southern Quark and remain fairly constant all the way to the Bothnian Sea. Westerlund et al. (2022) point out that boundary conditions are a major contributor to uncertainties in their model. Our study suggests that, in addition, turbulent mixing in the Southern Quark might not be accurately captured

by NEMO. We hypothesize our observations of mixing related to the overflow over the Southern Quark Sill as well as mixing across the halocline in a shallower region northwest of the sill (Muchowski et al. 2022b, under review) could partly explain mismatches. While Westerlund et al. 2022 compare CTD profiles from the Åland (station F64 in their Figure 1b) and the Bothnian seas (station F33 in their Figure 1b) with model output, we show additional data between those two stations, in the Southern Quark.


Oxygen measurements show a direct consequence of the observed mixing during the EL19 cruise where we have a reference station sufficiently far away from the sill (Fig. 1c): oxygen rates in the vicinity of the sill were higher along isohalines than those at the reference station south of the sill (Fig. 9).



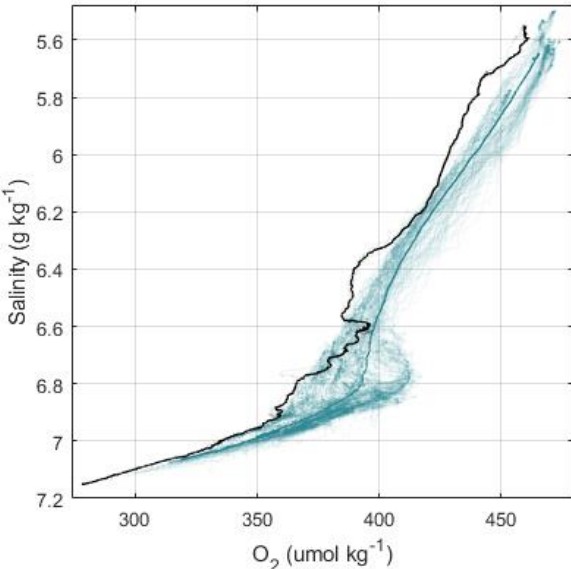


**Figure 9: Oxygen as a function of salinity, comparing 99 MSS casts collected near the Southern Quark Sill during EL19 (turquoise), with those from a reference station south of the sill (black) and their average (thick lines). Data from same MSS casts as shown in Figure 8 (turquoise and black, respectively).**

The average halocline depth near the sill (averaged over all profiles collected near the sill in each cruise) varies by more than 20 meters between EL19 and EL20 (Fig 8c) and by approximately 30 m between single profiles collected in different regions during EL20 (Fig. 8a,b). During EL19 we measured a drop of the halocline depth of 15 m within approximately 2 km distance in the vicinity of a stagnation wedge slightly north of the sill (MSS 263-264). The mixed layer depth south and north of the sill during this transect changes by 10 m (MSS 262 ca 15 m, MSS 264 ca 25 m).

Previous studies show a large variability and partial mismatch with observational data as well as when compared with other model results but lack an explanation. A previous study by Meier 2007, using the 3D eddy-permitting Rossby Center Ocean model estimated the halocline depth in the Southern Quark to be around 25 m (their figure 7), while Ahola 2021; Westerlund and Tuomi 2016; Håkansson et al. 1996 estimated the halocline depth to be between 50-80 m depth. Our observations show that the halocline depth in the Southern Quark has a high spatio-temporal variability of about 30 m depth within days and several kilometres, which may explain the mismatch mentioned above. Along the same lines as our observations, Elken et al. (2006) found that temporal counter-estuarine transport causes large changes in halocline depth (>20 m) and greatly increases turbulent mixing in the northern Baltic Proper, in particular near bathymetric features.

Changes in halocline depth in the northern Baltic Proper alter the amount of deep water and thereby the amount of salt, phosphorus, etc. that can enter the Åland Sea and eventually the Bothnian Sea. Additionally, the amount of saltier deep water





entering the Åland Sea from the northern Baltic Proper depends on water properties in the northern Baltic Proper. The described changes in halocline and surface mixed layer depth in space as well as over time, together with the large bathymetric variability in the Southern Quark will likely lead to mixing hotspots in different regions at different times and due to varying underlying mixing mechanisms. The fact that all above described processes occur on small scales of several hundreds of meters to several

kilometres, make in-situ observations as well as numerical simulations of the region cost and time intensive. In fact, established in-situ methods for observing dissipation rates are prone to fail in capturing the highly dynamic spatio-temporal developments in this region with drastically varying bathymetry due to their one-dimensional nature.

## 5. Conclusion

Average energy dissipation rates ($10^{-7}$-$10^{-6}$ W kg$^{-1}$) and salt flux rates ($10^{-2}$-$10^{-0}$ g m$^{-2}$s$^{-1}$) were increased by 3-4 orders of magnitude in reversing stratified overflow over a prototypical sill and through a connected deep valley compared to a reference station not directly impacted by the overflow. Our results highlight the importance of the Southern Quark for water exchange processes between the Northern Baltic Proper and the Bothnian Sea. High-resolution acoustic broadband midwater and multibeam bathymetry data were central in this study to observe and visualize the particularly dynamic and localized turbulent

mixing as well as to study its underlying mechanisms. Based on the latest EMODNET bathymetry of the Åland Sea, which shows a large variability in bathymetric features in the entire region, and our observational data, we conclude that the deep water (below the halocline) is likely continuously modified while flowing through the Åland Sea.

Based on the observational data presented here, and findings in Muchowski et al. 2022b (under review), we suggest that

enhanced mixing in the Southern Quark is caused mainly by two mechanisms: 1) stratified flow over rough seafloor bathymetry on the order of several hundreds of meters, that causes topographic wake eddies and to a smaller extent also breaking internal waves – this mechanism alone can modify the deep-water salinity by ~0.1 g kg$^{-1}$ (Muchowski et al. 2022b, under review). 2) blocking of the deep water by the Southern Quark Sill and the development of internal lee waves related to the overflow which travel upwards and break when encountering the halocline, shear instabilities due to opposing flow directions of the surface

layer with the water below, and hydraulic jumps. Hydraulic jumps are known to locally increase turbulent mixing by orders of magnitude (e.g. Farmer and Armi, 1998; Arneborg et al., 2017). Data presented in this study show that a hydraulic jump in the vicinity of a sill in the Southern Quark was present in >30% of our field measurements in 2019 and 2020, and that this is associated with strong vertical mixing at these time intervals. The observed mixing related to the steep bathymetric features depends greatly on the flow conditions, the stratification, halocline depth, and mixed layer depth. All these factors make a

general description as well as accurate numerical modelling of the exchange and water modification processes in the Åland Sea extremely challenging. Yet, mixing in the Southern Quark is likely not only important for the local conditions in the area but also broadly affects exchange processes between the Northern Baltic Proper and Bothnian Sea. The observed strong mixing

will likely impact the ventilation and residence time of the deep water, as well as oxygen, and nutrient concentrations in the Bothnian Sea, the surface water in the Northern Baltic Proper, and the larger-scale circulation in the entire Baltic Sea. For example, the observed mixing in the Southern Quark could impact cyanobacteria blooms in the Bothnian Sea as it affects the transport of phosphorous that originates in the Northern Baltic Proper and propagates northward through the Åland Sea (Rolff and Elfwing, 2015). An improved parametrization of mixing and basin exchange processes in the Northern Baltic Sea could be used together with the long-term monitoring data of sea-level changes south and north of the Aland Sea to create long-term estimates of the water exchange volume between the northern Baltic Proper and the Bothnian Bay, similar to the analysis of Major Baltic Inflows by Matthäus and Franck (1992).

**Author contribution**

JM lead the work and wrote most of the paper. MJ wrote parts of the introduction as well as the background on bathymetry section and contributed with feedback and editing on all other parts. LU, LA, and PH lead discussions on the interpretation of turbulence data and contributed with writing, editing and comments, BG and CH provided input on impacts and consequences for the Baltic Sea and contributed with feedback and comments, BG contributed in the section on sea surface height analysis, CS contributed with fruitful discussions, support and with writing and editing. All authors provided critical feedback and commented on the manuscript.

**Competing Interests**

The authors declare that they have no conflict of interest.

**Acknowledgements**

We thank Martin Sass and Toralf Heene (IOW, Germany) for technical support with the MSS and moorings. We thank Florian Roth, Ole Pinner, Emelie Ståhl, Jen-Ping Peng for helping with data collection. We thank captain Thomas Strömsnäs and crew of R/V Electra for their strong support.

**Data Availability**

Data and video supplements will be published and made accessible for downloading on the Bolin Centre Database website.



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
