# Peer review of "Observations of strong turbulence and mixing impacting water exchange between two basins in the Baltic Sea"

_EGUsphere, 2023_

## Author Comment (AC1)

RC1: 'Comment on egusphere-2023-920', Anonymous Referee #1, 03 Aug 2023 reply

The authors use shipboard acoustic and microstructure turbulence observations to study a dense overflow in the Baltic Sea. Results show how topographic lee waves propagate upwards into the halocline where they break and cause high levels of turbulent mixing.

This is a very nice paper and I don't have any major comments. There are a few minor points that would be good to be addressed before the paper is published. I trust that the authors will include my comments where they feel that changes are necessary and do not need to review a revised version before publication.

Minor comments:

14-15: This sentence reads maybe a bit generic?

Changed to: "Yet, current in situ observations often cannot capture the spatio-temporal development of dynamic and intermittent turbulent mixing related to overflows over rough bathymetry."

15: overflow -> overflows - *Changed*.

19: Possibly split sentence where you start talking about when data were collected.

The sentence on LL18 is now split into two and reads: "Our data were acquired during two oneweek cruises on R/V Electra in February-March 2019 and 2020. We collected high resolution broadband acoustic observations of turbulent mixing together with in situ microstructure profiler measurements, and current velocities from Acoustic Doppler Current Profilers.

20: Year incomplete. - Thanks. Fixed.

20: I had the impression that the fast timescale of the flow reversals was not previously known. If so it would be worthwhile to highlight the timescales here.

Thank you very much. We added on L23: "The observed flow reversed during both cruises on timescales of a few days."

23: You only speculate about submesoscale processes in this paper, this shouldn't be stated here as a result. - *We agree and deleted the sentence*.

23: leading -> led - *Was deleted when working on the comment above.*

24: Mixing -> Turbulent mixing? What specific quantity are you referring to here?

*Changed on L 26 to: "Dissipation rates of turbulent kinetic energy, vertical turbulent diffusivities and vertical salt flux rates were increased…" – also taking account comment from reviewer 2.*

29: The last sentence of the abstract reads a it generic, consider removing. -. *Removed*

33: You don't talk about your results (breaking lee waves cause turbulent mixing in the vicinity of the sill here). It remains a bit unclear why you refer to numerical models if you don't mention parameterizations. As stated above, I would move this into the discussion section and focus on the results of your study in this summary.

We assume this comment refers to the one above. We removed the sentence. We do talk about our results on LL 33 of the tracked changes version where we write: "We suggest ..."

43: Do you have a reference for the Bothnian Sea being well oxygenated?

Thank you for pointing this out. It's actually not correct to write that. The wording was too optimistic. We changed the sentence and added new references. LL 47 now read: "While the deep water of the Baltic Proper has become increasingly anoxic due to eutrophication and poor ventilation (Carstensen et al., 2014), oxygen concentrations in the deep water in the Gulf of Bothnia, which includes the Bothnian Sea and the Bothnian Bay north of it, have been decreasing in the last decades but have not reached anoxic conditions (Raateoja, 2013; Polyakov et al., 2022)."

We also adapted a related sentence on LL 71 of the tracked changes version which now reads: "Potentially related to this long-term feedback mechanism, it has been shown that phosphorus concentrations in the Bothnian Sea have increased in the last decades and are posing a risk of further oxygen depletion.":

54: lead -> led - Thanks, changed.

55: lead -> led - Thanks, changed.

62: "despite the fact that" or "despite anthropogenic, land-based sources having declined significantly"

Thank you. We chose your second suggestion. LL 68 reads now: "Despite anthropogenic, landbased sources of nutrients having declined significantly over the last decades,..."

64: over the last decades - *Changed, see reply to comment above.*

68-70: Consider moving this sentence to the end of the next paragraph or removing it as turbulent mixing and associated water mass transformation have not been discussed at this point.

We totally agree. Thank you very much. We moved the sentence to the end of the next paragraph, now on LL 93 in the tracked changes version of the manuscript.

77: remove comma after Bothnian Sea *Comma removed.*77: parentheses missing around 2007 *Parentheses added.*

86: Consider removing "Here," or replacing it with "In this paper,".

Replaced with "In this paper," on LL 96.

87: velocities -> velocity *Changed, see reply to comment below.*

87: You could be more precise and replace "mixing parameters" with "microstructure shear and temperature". *Changed, see reply to comment below.*

87: Remove one "and"

Merging the above four comments, the sentence was split in two and LL 96 in the tracked changes version reads now: "In this paper, we present observational data from the Southern Quark in the northern Åland Sea, including densely-spaced observations of current velocities, water column stratification, microstructure shear and temperature. Additionally, we present co-located acoustic observations from a broadband echosounding system that provide insights into the complex spatial structure of turbulent mixing at high resolution."

We also changed unprecedented to high.

95: Panel d is too small.

*We adapted Figure 1, also with regard to a comment from reviewer two. The updated map has a larger panel d.*

97: I suggest adding a reference for the red stars and removing the reference for selecting the area of your overview map.

The reference (now on LL 108 of the tracked changes version) belongs to the entire panel (b) – the choice of region and definition of the sills is taken from Westerlund et al. 2022, not only the red stars. We slightly changed the sentence to: "Overview over study area as shown in Westerlund et al. (2022): three major sills that separate the Northern Baltic Proper from the Bothnian Sea (black dashed rectangles), and positions of previous observations from monitoring stations (red stars)."

196: effect of mixing -> effect of turbulent mixing

We added turbulent on L 207 of the tracked changes version.

197: mixing data -> turbulence observations

Changed, now on L208 of the tracked changes version.

198 mixing -> turbulent mixing Added.

211: Change to published version. *Thank you. Changed.*

226: Possibly mention here that the "internal shear sensor" (I am still a bit puzzled by the name) is used to determine noise due to instrument vibration?

*We added on L237 of the tracked changes version: "..to monitor noise due to instrument vibrations"*

The internal shear sensor is basically a shear sensor with the airfoil probe replaced by a small weight. Using its inertia, this sensor is able to quantify lateral displacements (e.g., vibrations) of the profiler. We process the signal of this sensor the same way as the signal from the real shear sensors to derive a "pseudo dissipation rate" from it. This is the spurious dissipation that is detected by the external shear sensors due to high-frequency lateral profiler displacements, even if the water would be completely stagnant.

247: You mention a quality control algorithm but the data in Figs. 2a & 2b appear to show some noise. Maybe the quality control was only applied to the moored ADCP data?

Noise in the presented ship ADCP data could be due to the comparably short averaging time intervals. We chose to use those short averaging times nevertheless, as changes in current strength and direction appear on such small measuring timescales. We added the validation parameter of the ship ADCP data in the Supplementary material as Figure S3b next to the validation parameter of the mooring ADCP and mention this on L282 of the tracked changes version: ", corresponding validation parameter shown in Supplementary Fig. S3b"

**266: towards the north?**

We mean towards the south here. The surface flow was northward near the sill and then changed while we were going over the sill. When we measured north of the sill, the surface flow was southward. We rewrote the sentence also in reply to a comment from reviewer 2. It now reads on LL 278: "The simultaneously collected ship ADCP data reveal that in the first half of the almost 3-h long transect, the surface layer flow in the vicinity of the sill was mostly southward, as expected for an estuarine-type circulation, while at the very beginning of the transect as well as north of the sill, during the second half of the transect (at a distance of about 4.5 km from the start of the transect), the flow direction changed towards the south (Fig. 2a,b)."

270: Maybe it would be better to phrase this as "data collected along the transect are expected to show spatial and temporal variations" since you don't resolve this with one transect?

We rewrote the sentence, it now reads on LL 291: "Note that data collected along the transect are expected to show spatial as well as temporal variations, as the duration of the measurements was about three hours."

271: This sentence seems to terminate early? I can't quite make sense of it.

We rewrote the sentence. Now on LL 288 of the tracked changes version: "ADCP data (Fig. 3) from the mooring slightly north of the sill show that, during the time of the measurement, the main flow direction of the deep-water below 140 m depth was northwest. The main flow direction

of the intermediate water below the halocline but above 140 m depth (the near-surface layer was outside the range of this instrument) was northward."

277: How do you determine that some of the water is blocked by the sill? Looking at your (impressive!) acoustic observations in Fig. 4, I get the impression that most of the deep water is being drawn up to the sill height?

Reading your comments (including the ones below) and looking at TS plots from profiles just south of the sill compared to those on and just north of the sill, we agree that we actually cannot for sure distinguish between blocking and bottom water that gets diluted / is objected to entrainment while flowing over the sill. We got the impression that the lowest water was blocked because there is a comparably warm (yellow), salty (yellow), and oxygen-depleted (brown) patch of water just south of the sill visible in our MSS data in Figure 2c,d,e, respectively, at the very beginning of the transect and at about 180 m depth. We rewrote all parts related to blocking, see reply to your comment to L 324 below.

284: What is your definition of deep water here? Denser than a specific isopycnal? Some depth level?

We replaced "deep" by "bottom" water on L303 of the tracked changes version.

308: Black line -> The black line

Changed. Added "The" on L 328 of the tracked changes version.

324: Do you mean tilting density interfaces indicate that deep water is lifted up to the sill height? "due to" implies a physical process.

We agree that "due to" is the wrong wording here. We changed the sentence and split it in two. It now reads on LL 344: "South of the sill, eroding and tilting density interfaces indicate that the deep water is lifted up (white arrow in Fig. 4). North of the sill, biological single target scatterers become thin stretched-out lines, indicating that they are transported passively with the current and hence that the denser deep water falls down on the lee side of the sill."

Also, do eroding density interfaces imply that there is near-bottom turbulent mixing before the water reaches the sill crest? Could this explain the loss of oxygen and temperature minima you show in Fig. 2 without any blocking necessary?

Thank you very much. That's a very good point we've been blind to. Our data suggest indeed that there is mixing across the eroding density interface as well as near the sill which certainly can dilute the deepest water. Unfortunately, we do not have ADCP mooring data during the time when the transect shown in Fig. 4 was sampled to support or falsify the hypothesis of blocking. Bottomline, we agree with you that we actually cannot proof that there is blocking and phrased the text more carefully on LL 386: "This could be caused by a combination of partial blocking of the saline and warm deep water south of the Southern Quark Sill and its dilution due to entrainment and mixing when flowing over the sill." We also rewrote on LL 143: "All sills described above are most likely crucial for the exchange flow between the sub-basins. It is therefore important to understand their role in mixing of water masses" and LL 264: "Impacts of **the seafloor topography** and turbulent mixing on the properties of the water north of the sill are examined." and on LL 576: "2) **potential** blocking of the deep water by the Southern Quark Sill and the development of internal lee waves related to the overflow..."

| 356: 3degC or 3K | No idea where the C went, it's there now. Thank you. |
|------------------|------------------------------------------------------|
| 358: 6degC or 6K | No idea where the C went, it's there now. Thank you. |

380: The map inset is too small.

We made panels (a) and (b) larger and updated Figure 5 with enlarged inset.

365: Reiterating my points above, how do you determine that this is due to blocking and not mixing and entrainment?

See our reply above. We agree and rewrote the text.

391: I could not figure out how to retrieve the videos. In general I would like to suggest to treat the videos as what they are - supplementary material - and to include a figure that summarizes what you show in the video. Lines 399 to 407 could refer to one figure with several panels showing the acoustic observations at the times discussed in the text?

Good idea =) We added Figure 6, showing 12 snapshots, 4 from each video in each row, of echograms that show what we describe on lines 426-434. We added references to the Figure accordingly and renamed all following Figures and their references by +1. Nevertheless, we hope you get to see the videos.

423: shown below -> discussed below

Changed to "discussed below".

449: Individual profiles in this figure are shown with such faint lines that are impossible to read. Consider increasing their thickness or not showing them at all.

We increased the thickness of individual profiles in panels (a)-(c), removed single profiles in panels (d)-(f), and updated Figure 8 – which now became Figure 9 because we added the new Figure 6.

464: You are still in section 4.2 and the jump into the discussion with this paragraph seems rather abrupt here. Why not reserve section 4 for your results and move the discussion into the next section?

We agree that there is a bit of a jump. Yet, we would prefer to keep the paragraph on the Westerlund et al. 2022 model study in section 4 (Results and Discussion). As we have several small results it seems more convenient to discuss them right away where we present them in

order to avoid repetition. In section 5 we only write a conclusion and try to not discuss results anymore.

481: It would be good to state here why higher oxygen near the sill is a consequence of mixing - I am guessing it's because oxygen-rich waters from the north are entrained?

Thank you very much. We included your suggestion. The paragraph on oxygen, LL 525 in the tracked changes version, reads now: "Oxygen measurements show a direct consequence of the observed mixing during the EL19 cruise where we have a reference station sufficiently far away from the sill (Fig. 1c). Due to entrainment of oxygen-rich waters from north, oxygen rates in the vicinity of the sill were higher along isohalines than those at the reference station south of the sill (Fig. 10). "

495: This sentence reads very generic, consider removing or being a bit more specific.

We rewrote the beginning of the sentence on LL 538 of the tracked changes version to: "The following studies show a large variability and partial mismatch with observational data as well as when compared with other model results but lack an explanation. A study by Meier 2007...".

This and the following paragraph may benefit from being moved into a discussion section.

We would prefer to keep results and discussion combined in section 4. See also our reply to comment 464 above.

505: change etc to something more specific, or at least "and other tracers".

We changed etc. to "and other dissolved substances" on L 549 of the tracked changes version.

510: make -> makes Thanks. We added an "s" on L554.

510: "established in-situ methods for observing dissipation rates" - you probably mean microstructure turbulence observations? If so good to spell this out.

*We followed your suggestion and changed the wording to "microstructure turbulence observations" on L 555.*

511: "prone to fail" seems a bit harsh - you show with this paper that there is a lot to be learned even when undersampling in space and time.

We rewrote the sentence on LL 555 of the tracked changes version to: "In fact, microstructure turbulence observations often cannot resolve the highly dynamic spatio-temporal developments in this region with drastically varying bathymetry due to their one-dimensional nature.

515: Maybe it's just me but  $10^{-0}$  just looks a bit odd... Changed to "1" on L560.

516: in a reversing *We added "a" on L561.*

522: Your reference profiles south of the sill in Fig. 8d show that there are regions with relatively small turbulent mixing! Doesn't this contradict your statement of continuous modification?

We rewrote the second half of the sentence on LL 567 to: "we conclude that the deep water (below the halocline) is likely modified in several locations along its way through the Åland Sea."

525: Wake eddies have not been discussed prior to this. It may be helpful to summarize your GRL paper in one or two sentences when you refer to it for the first time.

We agree and added a sentence on LL 570 of the tracked changes version. The beginning of the paragraph reads now: "Muchowski et al. (2022b) investigated diapycnal mixing in the shallower, north-western part of the study region (yellow dots in Fig. 1c), including the amount of turbulent mixing across the halocline observed in this region, and discussed potential underlying mixing mechanisms. Based on the observational data presented here, and the findings in Muchowski et al. (2022b), we suggest..."

533: Consider starting a new paragraph at "The observed mixing".

*We followed your suggestion and started a new paragraph in L 582 in the tracked changes version.*

Citation: https://doi.org/10.5194/egusphere-2023-920-RC1

Additionally, we included a new reference to Eilola and Stigebrandt for Baltic Sea circulation and replaced with it the previously given reference to Hakanson and Bryhm 2008 as this one is less original. LL 211 of the tracked changes version now reads: "southward flowing water, which becomes part of the surface water in the Northern Baltic Proper (Hela, 1958; **Eilola and Stigebrandt, 1998**; Markus Meier et al., 2006)."

---

## Author Comment (AC2)

**• RC2: 'Comment on egusphere-2023-920', Anonymous Referee #2, 26 Aug 2023 reply**

In the paper by Muchowski et al. new observations around a sill in the Southern Quark region (Baltic Sea), i.e. the area connecting the Northern Baltic Proper with the Bothnian Sea, are presented. The new dataset is massive and comprehensive as it includes velocity and hydrographic data but also microstructure measurements as well as high-res acoustic observations of turbulent mixing. Results show that turbulent diffusivities, dissipation and vertical flux rates are very large and about 3-4 orders of magnitude bigger near the sill with respect to reference unperturbed stations. Such a strong mixing is thought to result from hydraulic jumps and stationary lee waves and shown to affect also oxygen values, impacting the ventilation and residence times of the deep layers in the region.

The paper is well written and organized and fits well the scope of the journal. I have only a major concern related to the large diffusivity values shown in Figure 8 which are reported to reach  $10^{-1}-1$  m2/sec in the deeper layers and even be larger than 1 at about 160-m of depth. I urge the authors to discuss these large values and compare with those observed in other areas. Can this be related to the choice of a constant mixing efficiency?

This is a very good point and was indeed misleading. The high diffusivities were related to the bottom boundary layer where assumptions of isotropy and a constant mixing efficiency break down. To avoid the problem, we removed the lowermost 10 m of data in all MSS profiles and we entirely removed two MSS profiles (MSS 131 from EL19 and MSS 143 from EL20) which have outliers in dissipation rates slightly above 10 m from the seafloor. We updated the corresponding information in the Supplementary material tables ST3-ST4, updated Fig. 8 (which now became Fig. 9) including the figure caption, and added this information on LL491 of the tracked changes version: "For the calculation of dissipation rates, vertical turbulent diffusivities and salt fluxes, we removed the lowest 10 m of data close to the seafloor."

A process of revisions is suggested to address also the following minor concerns:

1. L24-25: the large values of mixing should be reported in the abstract

We included the values in the abstract on LL 28 of the tracked changes version which now reads: "Dissipation rates of turbulent kinetic energy, vertical turbulent diffusivities and vertical salt flux rates were increased by 3-4 orders of magnitude in the entire water column in the vicinity of the sill compared to reference stations not directly influenced by the overflow with average dissipation rates near the sill between  $10^{-7}$  and  $10^{-6}$  W kg-1, average vertical diffusivities of  $0.001 \text{ m}^2 \text{ s}^{-1}$  in the halocline and up to  $0.1 \text{ m}^2 \text{ s}^{-1}$  below the halocline, and average vertical salt flux rates around  $0.01 \text{ g m}^{-2} \text{ s}^{-1}$  in the halocline and between  $0.1 \text{ and } 1 \text{ g m}^{-2} \text{ s}^{-1}$  below the halocline."

2. Fig1: please add a rectangle in panel a to show the close-up of panel b like it is done in b with the yellow rectangle for panel c.

We added a yellow rectangle in (a).

Many dots in panel c are barely visible also for the color choices (green and turquoise over a blue or light-blue bathymetry). I suggest to have panel c and its inset (panel d) way larger and below panels a and b

We incorporated the suggested changes and updated Figure 1.

3. L135-141: Not sure these lines are relevant. Why is the bedrock geology important for this study?

It simply explains why we have such a dramatic seafloor topography in the first place, which is a premise for the mixing we address. We have removed some details from the paragraph and added a sentence, putting it better into context. The paragraph on LL 146 reads now:

"The dramatic seafloor morphology of the Southern Quark is to a large extent inherited from the underlying bedrock geology. The Southern Åland Sea basin and Lågskär Deep were formed from a tectonic depression underlain by 1.0-1.6 Ga old sandstone, while the rough seafloor areas surrounding Åland as well as the islands themselves, are predominantly comprised of even older crystalline bedrock, i.e. to a large extent the famous Rapakivi granite (EMODnet Geology) (Beckholmen and Tiren, 2009). The steep ridge in the Southern Quark mentioned above is proposed to be composed of dolerite, which is a resistant magmatic rock (Beckholmen and Tirén, 2009). The Southern Quark is a typical example of how the underlying geology often serves as the foundation for the rough seafloor, implying that the general geology may provide valuable insights into identifying critical regions for turbulent mixing, in the Baltic Sea or elsewhere."

4. L151-152: very true but also consider that ocean models need to have enough horizontal resolution to fully resolve the bathymetric features present in the DBM dataset and that it is usual practice in some ocean models (e.g. sigma models) to smooth bathymetry

Yes, this is true. It underlines the importance of high-resolution bathymetry data to parametrize mixing based on bathymetry roughness. The influence of the bathymetry on mixing and drag should, ideally, still be included.

5. L161: what kind of mixing values (i.e. diffusivity values) are expected to sustain such an estuarine-type circulation? How do they compare with those observed in this study?

Thank you. That's a very good idea to include those values. We added on LL 496 "We find that energy dissipation  $(10^{-7} - 10^{-6} W \text{ kg}^{-1})$ , turbulent vertical mixing  $(0.001 - 0.1 \text{ m}^2 \text{s}^{-1})$  and vertical salt flux rates  $(0.01 - 1 \text{ g m}^{-2} \text{s}^{-1})$  are increased by two to four orders of magnitude in the entire water column near the Southern Quark Sill compared to reference stations south of the sill, during both cruises EL19 and EL20 (Fig. 9d-f)."

and added the following paragraph on LL 500: "To maintain the general estuarine-type circulation in the Baltic Sea, an estimated mean diapycnal salt transport (from the deeper water across the pycnocline into the surface water) of 30 kg m-2 a-1 is needed (Reissmann et al. 2009). This equals a vertical salt flux of approximately 0.001 g m-2 s-1. The in this study measured vertical salt transport in the halocline is in both regions, near the sill as well as in the north-western shallow part of the study region, with around 0.01 g m-2 s-1 an order of magnitude larger than this. It is important to note that Reissmann's estimate includes vertical transport during Major Baltic Inflows as well as all other sources of mixing."

6. L264-270: really confused mainly by the text here. It looks to me that at the beginning the ship ADCP data show northward velocities and not southward as written in the text while after 1.5 hours velocities are back towards the north and not the south. Please check, clarify and rephrase

We apologize for the confusion and admit the somewhat sloppy formulation here. At the very beginning of the transect the flow was indeed already northward. We rewrote the text starting on LL 278, trying to be a bit more precise and clear: "The simultaneously collected ship ADCP data reveal that in the first half of the almost 3-h long transect, the surface layer flow in the vicinity of the sill was mostly southward, as expected for an estuarine-type circulation, while at the very beginning of the transect as well as north of the sill, during the second half of the transect (at a distance of about 4.5 km from the start of the transect), the flow direction changed towards the south (Fig. 2a,b). Corresponding wind data from SMHI station Örskär (60.5256 °N, 18.3729 °E) in the vicinity of the study region reveal a shift in wind direction and drop in wind speed during the time of the measurement that coincides with the change in surface water current direction in the second half, about 1.5 hours into the transect (Supplementary Fig. S1).

7. L271-275: confused again as ADCP data are showing northward velocities also below the halocline but above 140m. Is this consistent with an estuarine-type circulation? If yes please explain better

The very bottom water below 140 m flows north**west**ward and the intermediate water below the halocline down to 140 m flows northward.

Reviewer 1 commented on the sentence as well. We rewrote on LL 289 of the tracked changes version to: "The main flow direction of the intermediate water below the halocline but above 140 m depth (the near-surface layer is outside the range of this instrument) was northward, as expected for an estuarine-type circulation."

8. Figure 2, panels c-f: please explain how is interpolation performed

We manually perform a linear interpolation to interpolate all data plotted in one panel on the same z axis grid and then use Matlab function contour to plot the data - which interpolates linearly between data points in 2D.

We changed Figure caption on LL 313 of the tracked changes version to: "(c)-(f) **Linear** interpolation of MSS 212-230, excluding casts 221 and 224 which were aborted. (c) Conservative temperature, (d) absolute salinity, (e) oxygen concentration, (f) energy dissipation rate. Black isopycnals **plotted with Matlab function contour** at intervals of 0.05 g kg-1, grey isopycnals at 0.01 g kg-1."

9. L325: why not saying simply white arrow instead of bright grey? Initially I got lost try to find a different grey arrow

We changed the caption and text to "white arrow".

10. L334: how can you be sure to say that below 100 m the echoes are due to eddies or large overturns?

We rewrote the sentence on LL 356 of the tracked changes version to: "In the deeper regions below 100 m, however, turbulence microstructure related to large overturns, eddies, and internal-wave oscillations is ubiquitous in the echogram (Fig.4, seen as more fuzzy scattering without clear edges) as well as in the microstructure data (Fig. 2c-e), causing increased mixing all the way down to the seafloor (Fig. 2f). Further away from the sill, increased mixing rates in the deep water are likely also influenced by the steep walls on both sides of the valley."

11. L339-340: what does "where there is little temperature and salinity microstructure" mean? Do you mean gradient?

*We rewrote the sentence on LL 362 of the tracked changes version to: "where temperature and salinity gradients are small,"*

12. L374: the dark blue dots in Fig.5f are barely visible

We added white circles around all dots in Figure 5f to make especially the dark blue dots better visible. A revised version of Figure 5 is now included in the manuscript. We further noticed that we hadn't mentioned the positions of the moorings in the caption of Figure 5 and added it now.

13. Figure 5: what about the two relative maximum temperature values for the black line across the 50-m depth? Intrusion of intermediate waters?

We agree and added on L395: "Furthermore, MSS profiles 229 and 230 have two pronounced local temperature maxima just above and below 50 m depth. We hypothesize that those could be intrusions of intermediate waters."

14. L409-425: the first and last lines of this paragraph seem to me to contradict each other as currents are indeed stronger and more persistent during EL19 and thus not comparable during both cruises. Or am I not grasping something here?

You're right! We changed the beginning of the paragraph on LL 441 of the tracked changes version to: "Mooring ADCP data show that during EL19, a flow reversal occurred only once,

from the dominant northward direction to southward flowing currents on the second day of the cruise. In contrast, during EL20,..."

15. Figure 6: It looks there is an argument here that the wind sets up a ssh difference responsible for a barotropic signal. Wouldn't be possible to filter out the pressure-based signal to show the residual (estuarine-type?) circulation? Will a EOF-based approach work?

An EOF based approach would be a good idea to separate different modes of variability. However, we find that it is difficult to conclude much on barotropic vs. baroclinic motions from our ADCP mooring data set as it does not cover the upper 50 m of the water column.

16. Figure 7: just to point out that the velocities are indeed weaker but also less barotropic

Good point to mention. We added the information on LL 456 of the tracked changes version which now reads: "Overall currents north of the sill changed direction more frequently, were weaker, and less barotropic during EL20 compared to EL19 (compare Fig. 7 with Fig. 8a)."

**17. Figure 8: impressed by the large numbers here. Why do kz values increase below 200 m for the unperturbed profiles?**

Thank you for pointing this out. We added on LL 492 of the tracked changes version: "The increase in vertical diffusivities below 200 m depth in the EL19 reference measurements (Fig. 9e, black lines) is likely related to the decrease in buoyancy frequency (Fig. 9c, black lines), as the measured dissipation rates are close to the noise level of the profiler."

Why are the yellow lines useful or, in other words, what is the point of reporting values for the north-western part? For comparison?

Because we talk about water being modified along its way through the Aland Sea in many different areas. The sill that we discus is one big part but we think it is important to note that the sill is just part of the puzzle and there are many other mixing hotspots in the region. We added the yellow lines to compare this other region to the sill region. Furthermore we find it interesting to see how much T,S profiles, including halocline and mixed layer depth change over just a couple kilometers.

**18. L515: please report here (and also in the abstract) also the diffusivity values**

On LL 26 in the abstract we now included all values: "Dissipation rates of turbulent kinetic energy, vertical turbulent diffusivities and vertical salt flux rates were increased by 3-4 orders of

magnitude in the entire water column in the vicinity of the sill compared to reference stations not directly influenced by overflow with average dissipation rates near the sill between  $10^{-7}$  and  $10^{-6}$  W kg-1, average vertical diffusivities of  $0.001 \text{ m}^2 \text{ s}^{-1}$  in the halocline and up to  $0.1 \text{ m}^2 \text{ s}^{-1}$  below the halocline, and average vertical salt flux rates around  $0.01 \text{ g m}^{-2} \text{ s}^{-1}$  in the halocline and between  $0.1 \text{ and } 1 \text{ g m}^{-2} \text{ s}^{-1}$  below the halocline."

On L 560 we also included the values of vertical diffusivities: "Average energy dissipation rates  $(10^{-7}-10^{-6} W \text{ kg}^{-1})$ , vertical diffusivities  $(0.001-0.1 \text{ m}^2 \text{ s}^{-1})$  and salt flux rates  $(0.01-1 \text{ g m}^{-2} \text{s}^{-1})$  were increased by 3-4 orders of magnitude in a reversing stratified overflow..."

Citation: https://doi.org/10.5194/egusphere-2023-920-RC2

Additionally, we included a new reference to Eilola and Stigebrandt for Baltic Sea circulation and replaced with it the previously given reference to Hakanson and Bryhm 2008 as this one is less original. LL 211 of the tracked changes version now reads: "southward flowing water, which becomes part of the surface water in the Northern Baltic Proper (Hela, 1958; **Eilola and Stigebrandt, 1998**; Markus Meier et al., 2006)."